# FACEGANS: STABLE GENERATIVE ADVERSARIAL NETWORKS WITH HIGH-QUALITY IMAGES

**Jiayu Li, Zheng Zhan, Caiwen Ding, Yanzhi Wang**
Department of Electrical Engineering & Computer Science
Syracuse University
Syracuse, NY 13244, USA
`{jli221, zzhan03, cading, ywang393}@syr.edu`

## ABSTRACT

Generative Adversarial Networks (GANs) have shown impressive performance in producing images highly similar to original dataset under unsupervised learning. However, the losses of discriminator and generator are highly fluctuated, which affects the quality of fake images produced by the generator. In this work, we propose Face Generative Adversarial Networks (FaceGANs). Compared to the conventional GANs, our new structure can stabilize the loss fluctuation of discriminator and generator. It also improves the capabilities of generator and discriminator. In order to fully investigate FaceGANs, we compare the performance of FaceGANs with Deep Convolution Generative Adversarial Network (DCGANs) on the Celeba dataset. Experimental results show that our FaceGANs structure can fast generate images with better quality than DCGANs in a facial reconstruction.

## 1 INTRODUCTION

Goodfellow et al. (2014) proposed GANs, which is a neural network composed of two parts — discriminator and generator. The generator uses noise data to create potential samples to deceive the discriminator. In the meantime, the discriminator should try to improve the discrimination in the learning. When it comes to a balance that the generated images cannot be recognized by the discriminator, the produced images are highly similar to original dataset. Radford et al. (2015) showed us DCGANs, a strong candidate for unsupervised learning. DCGANs have shown the applicability of general image representations.

However, we find that the loss of discriminator and generator are not stable enough and highly fluctuated in GANs and DCGANs. The situation will affect the quality of images from a generator. In this work, we introduce FaceGANs, a new structure of GANs and employ an improved loss function. Under the same frequency of training, experimental results show that our FaceGANs structure can generate images with better quality than DCGANs in a facial reconstruction.

### 1.1 RELATED WORK

Several works considered how to improve quality of images from GANs or DCGANs. Mao et al. (2017) found loss function may lead to the vanishing gradients problem during the learning process. They employed the least squares in loss function for the discriminator from the view of minimizing the Pearson divergence.

Metz et al. (2016) proposed unrolled generative adversarial networks. This networks aimed to stabilize GANs. The Optimal discriminator is ideal but in-feasible in practice in the generators objective. The discriminator of GANs is often unstable and leads to poor solutions. This innovation allows training to be adjusted between using the discriminators.

Berthelot et al. (2017) found a method to balances the generator and discriminator during training. Additionally, it provides a new approximate convergence measure, fast and stable training. Also, Dosovitskiy & Brox (2016) computed distances between image features extracted by deep neural networks to generate high-resolution images.

## 2 FACEGANS AND INNOVATION

Goodfellow et al. (2014) proposed the generator and discriminator with enough capacity will reach a point at which both cannot be improved after several steps of training. We build a new structure to balance generator and discriminator. More specifically, the generator and discriminator have similar

capacity and therefore, the losses of generator and discriminator can be stabilized and close to each other.

## 2.1 ARCHITECTURE OF FACEGANS

Differentiated from CoGAN (Liu & Tuzel (2016)) and SGAN (Huang et al. (2017)), we proposed a symmetric structure in FaceGANs, implementing a balanced structure between generator and discriminator. For the generator (shown in Figure 1), we use two fully connected layers in the first two low levels and four deconvolutional layers in the deep layers. The first fully connected layer is responsible for extending the data into a large size and the second fully connected layer abandons some redundant data and shrinks input into a proper size. Between each layer, we use batch normalization and Leaky ReLUs. Batch normalization (Ioffe & Szegedy (2015)) has two-fold advantages: faster learning and higher overall accuracy. The Leaky ReLUs generates a small, non-zero gradient if not activated. The FaceGANs architecture allows us to use a higher learning rate, significantly boosting the training speed.

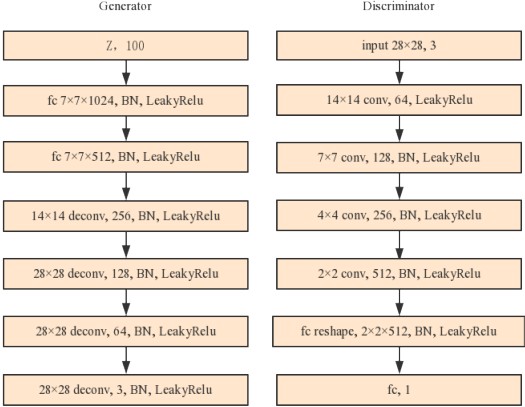

Figure 1: Architecture of generator and discriminator.

For the discriminator (shown in Figure 1), we use four convolutional layers in the first four levels and two fully connected layers in last two layers, which is an opposite structure to the generator. Between each layer with Leaky ReLUs, we employ batch normalization except the first convolutional layer. The purpose of using opposite structures in generator and discriminator is to balance the capability of discriminator and generator, which addresses the unbalanced problem between generator and discriminator Metz et al. (2016) proposed before. Through using the architecture, we improve the adaptive capability of the networks.

## 2.2 LOSS FUNCTION OF FACEGANS

We use $G(z)$ to represent the loss of generator and $D(x, z)$ to represent the loss of discriminator. $D(G(z))$ shows results from the discriminator's judgment about the output from the generator. However, the value of $D(G(z))$, $(1 - D(G(z)))$ and $D(x)$ will be disperse. The $sigmoid$ function can restrict the value from 0 to 1 and also provide the probability of $D(G(z))$, $D(x)$ and $(1 - D(G(z)))$. Therefore, we use $sigmoid$ function into the loss function. We aim to maximize the $G(z)$. If the generator is better than the discriminator, the discriminator will recognize fake images from the generator as real images and $D(G(z))$ will become larger. The final loss of $G(z)$ will become larger. We define the following loss function for the generator:

$$\max G(z)_{loss} = \frac{1}{N} \times \sum - \log(sigmoid(D(G(z)))) \tag{1}$$

For the loss of discriminator, the purpose is to maximize the loss function. If the discriminator has enough training, $D(G(z))$ will become smaller because discriminator can recognize many fake images correctly, and $(1 - D(G(z)))$ will become bigger. Meanwhile, $D(x)$ become larger than before. The following format shows the definition of loss function of discriminator:

$$\max D(x, z)_{loss} = \frac{1}{N} \times \sum - [\log(sigmoid(D(x))) + \log(sigmoid(1 - D(G(z))))] \tag{2}$$

## 3 EXPERIMENT

Based on Celeba dataset and MNIST dataset, we compare the performance of FaceGANs with DC-GANs.

### 3.1 LOSS COMPARISON

Using the Celeba dataset, FaceGANs reaches the relative balance between generator and discriminator at around the 80th iteration whereas DCGANs need more iterations. And the loss distance between generator and discriminator in FaceGANs is smaller than 1 as shown in Figure 3, while the loss distance between generator and discriminator in DCGANs is larger than 1 and the loss of discriminator is always higher than generator as shown in Figure 2. Therefore, The loss of Face-GANs is more stable and less fluctuated. Additionally, during the initialization of FaceGANs, the discriminator lacks enough training, which leads to $D(G(z))$ larger (the discriminator recognizes fake images from the generator as real images) and decreases $(1 - D(G(z)))$. Thus, based on Equation (1) and (2), the loss of generator is larger than the loss of discriminator during the initialization. With more training, discriminator and generator become more adversarial. The distance of loss from discriminator and generator becomes smaller, which means their capacities are close. Therefore, the FaceGANs will generate better fake images with less training.

For MNIST dataset, at about the 60th iteration, the capability of generator and the capability of discriminator in FaceGANs can be adversarial, which means they can help each other improve by training and keep balance in confrontation.

Thus, these experimental results satisfy the theory, which was proposed in (Goodfellow et al. (2014)), that if the generator and the discriminator have enough capacity after several steps of training, they will reach a point at which both cannot improve because of $p(g) = p(data)$. The discriminator is unable to differentiate between the two distributions, i.e., $D(x) = \frac{1}{2}$.

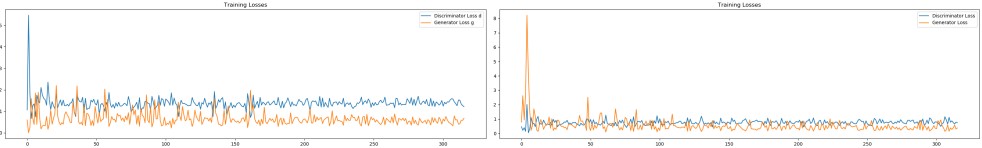

Figure 2: loss from DCGANs for Celeba dataset.          Figure 3: loss from FaceGANs for Celeba dataset.

### 3.2 IMAGES COMPARISON

Under the same frequency of training (networks output results after 3160th iteration), the generator of FaceGANs generates images with better quality and closer to real pictures than what DCGANs does (shown in Figure 4 and Figure 5).

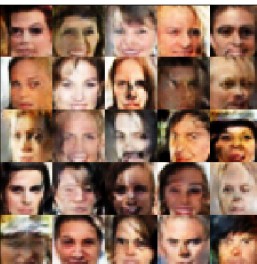     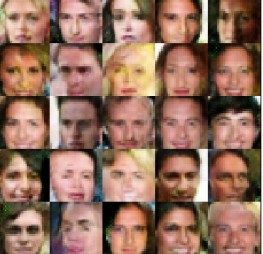

Figure 4: Result from DCGANs,          Figure 5: Result from FaceGANs,
3160th iteration                                3160th iteration

## 4 CONCLUSION AND FUTURE WORK

We propose a new structure and loss function of FaceGANs. The performance of FaceGANs on the Celeba dataset and MNIST dataset proves its superiority over DCGANs. The FaceGANs can reconstruct human face faster with less training iterations. In the future, we will apply FaceGANs to SVHN and CIFAR10 datasets. Moreover, we will employ a metrics method like the Inception score to evaluate the pictures produced by FaceGANs.

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
