# OpenReview forum: "FaceGANs: Stable Generative Adversarial Networks with High-Quality Images"
_ICLR.cc/2018/Workshop — Reject_

### Official Review · AnonReviewer2 · 2018-02-22
**No novelty.**

**Rating:** 1
**Confidence:** 5

**Review:**

This method, inexplicably called FaceGANs (despite not being limited to the domain of faces), appears to propose an innovation in the network architecture used for the generator and the discriminator that reduces fluctuations in the losses being optimized. The contribution appears to be a particular network architecture with nothing novel or interesting about it.

This seems to make a lot of unstated assumptions, such as that the learned model is "good" if the values for the losses are similar (which is necessary at equillibrium but not a sufficient condition for it), and that a more stable learning curve is desirable, and that faster convergence necessarily means the model is better.

The losses appear to be similar (or identical) to the classic GAN losses. Where the original paper presumed the range of D to be (0, 1), the authors here seem to find it necessary to explicitly introduce the sigmoid function. I am assuming that the second term of (2) is an error, and that they mean to say log(1 - sigmoid(....))) and not log(sigmoid(1 - ...)) which doesn't make any sense.

Evaluation involves qualitative inspection of plots and generated samples. The authors claim their samples are "better quality and closer to real pictures than what DCGAN does". I find this claim uninteresting and, besides that, impossible to judge.

The paper is confusingly written and hard to follow. Many statements are unclear, vague, or simply don't follow. For example:

- "DCGANs have shown the applicability of general image representations." Applicability to what? What representations?
- "The discriminator of GANs is often unstable" -- what does this mean? How can a neural network itself be unstable?
- "The situation will affect the quality of images from a generator" -- Stated without citation, explanation or justification
- "This innovation allows training to be adjusted between using the discriminators." -- This is only in the discussion of related work but it remains unclear what this means.
- "With more training, discriminator and generator become more adversarial." I don't have any idea what this means.
- "The distance of loss from discriminator and generator becomes smaller, which means their capacities are close." There is no reason to believe that this follows.

---

### Official Review · AnonReviewer3 · 2018-03-09
**assumptions without backing, lack of novelty and justification**

**Rating:** 4
**Confidence:** 5

**Review:**

The paper starts off with:

> "However, we find that the loss of discriminator and generator are not stable enough and highly fluctuated in GANs and DCGANs. The situation will affect the quality of images from a generator."

There is no backing for this statement, and afaik this is simply not true. The loss of a GAN (Goodfellow 2014)  is not correlated with sample quality.

The paper pitches an architectural rule that make sure that "the generator and discriminator have similar capacity and therefore, the losses of generator and discriminator can be stabilized and close to each other."

The rule is that you find inverted equivalents of generator architectures for discriminator architectures. For example Conv in discriminator is ConvTranspose in generator, with same capacity.
This is also a property of the DCGAN architecture that they put together as a baseline, so I am not sure what the novelty is.

Finally, I find the results section weak, even for a workshop paper. They show loss curves and show that loss "gaps" are more subdued with FaceGANs, but I fail to see why this is significant or why it's correlated with sample quality.

The Images Comparison is likely cherry-picked to whatever epoch is suited for FaceGANs, because the original DCGAN paper has faces of better quality than what they report.

---

### Official Review · AnonReviewer1 · 2018-03-10
**Insufficient empirical validation of proposed changes**

**Rating:** 3
**Confidence:** 5

**Review:**

The paper proposes an architectural modification to a DCGAN. The changes include using LeakyRelu and fully connected layers in the generator. For a loss function, standard Non-Saturating GAN (NS-GAN) is used though the language of section 2.2 does not make it clear that this is the same approach used in the original GAN paper. For a quantitative metric, the paper reports NS-GAN generator and discriminator losses. This is known to not be a meaningful loss metric for convergence as discussed in Section 4.2 of Wasserstein GAN (Arjovsky et al. 1701.07875) and is not an established evaluation metric. The only other experimental result is a snapshot of samples from a DCGAN and the paper's model at 3160 updates. As a reviewer I can maybe visually distinguish the two a little bit, but there is no significant difference and the results are far worse than those reported in the original DCGAN paper ~2.5 years ago (see Figure 10 from Radford et al. 1511.06434). Given the heuristic motivation of the proposed changes, thorough empirical work is needed to demonstrate a valuable contribution has been made. Currently the paper does not convincingly do this.

---

### Decision · Program_Chairs · 2018-03-20
**ICLR 2018 Workshop Acceptance Decision**

**Decision:**

Reject

**Comment:**

Based on the reviews, this paper has not been accepted for presentation at the ICLR workshop. However, the conversation and updates can continue to appear here on OpenReview.